# Accelerometer-measured 24-hour movement behaviours over 7 days in Malaysian children and adolescents: A cross-sectional study

**Sophia M. Brady[1], Ruth Salway[1,2], Jeevitha Mariapun[3], Louise Millard[2], Amutha Ramadas[4], Hussein Rizal[4], Andy Skinner[2], Chris Stone[5], Laura Johnson[1,2‡], Tin Tin Su[4,6‡], Miranda E. G. Armstrong[1‡]***

**1** Centre for Exercise, Nutrition & Health Sciences, School for Policy Studies, University of Bristol, Bristol, United Kingdom, **2** Department of Population Health Sciences, Bristol Medical School, University of Bristol, Bristol, United Kingdom, **3** Clinical School Johor Bahru, Jeffrey Cheah School of Medicine and Health Sciences, Monash University Malaysia, Malaysia, Malaysia, **4** Jeffrey Cheah School of Medicine and Health Sciences, Monash University Malaysia, Malaysia, Malaysia, **5** Integrative Cancer Epidemiology Programme, MRC Integrative Epidemiology Unit, and School of Psychological Science, University of Bristol, Bristol, United Kingdom, **6** South East Asia Community Observatory (SEACO), Jeffrey Cheah School of Medicine and Health Sciences, Monash University Malaysia, Malaysia, Malaysia

‡ LJ, TTS and MEGA share the joint senior authorship on this work.
* miranda.armstrong@bristol.ac.uk

**Data Availability Statement:** Our data availability statement is as follows: Data cannot be shared publicly because of confidentiality and ethical reasons. De-identified data are available and can be

## Abstract

### Background

Quantifying movement behaviours over 24-hours enables the combined effects of and inter-relations between sleep, sedentary time and physical activity (PA) to be understood. This is the first study describing 24-hour movement behaviours in school-aged children and adolescents in South-East Asia. Further aims were to investigate between-participant differences in movement behaviours by demographic characteristics and timing of data collection during Ramadan and COVID-19 restrictions.

### Methods

Data came from the South-East Asia Community Observatory health surveillance cohort, 2021–2022. Children aged 7–18 years within selected households in Segamat, Malaysia wore an Axivity AX6 accelerometer on their wrist for 24 hours/day over 7 days, completed the PAQ-C questionnaire, and demographic information was obtained. Accelerometer data was processed using GGIR to determine time spent asleep, inactive, in light-intensity PA (LPA) and moderate-to-vigorous PA (MVPA). Differences in accelerometer-measured PA by demographic characteristics (sex, age, ethnicity, socioeconomic group) were explored using univariate linear regression. Differences between data collected during vs outside Ramadan or during vs after COVID-19 restrictions, were investigated through univariate and multiple linear regressions, adjusted for age, sex and ethnicity.

### Results

The 491 participants providing accelerometer data spent 8.2 (95% confidence interval (CI) = 7.9–8.4) hours/day asleep, 12.4 (95% CI = 12.2–12.7) hours/day inactive, 2.8 (95% CI =

freely requested from the South East Asia
Community Observatory, Monash University
Malaysia Institutional Data Access at "mum.
seaco@monash.edu" for researchers who meet the
criteria for access to confidential data. For more
information, please refer to https://www.monash.
edu.my/seaco/research-and-training/how-to-
collaborate-withseaco.

**Funding:** This work was supported by the Medical
Research Council, grant reference MR/T018984/1
and the Ministry of Higher Education/UK-MY Joint
Partnership on Non-Communicable Diseases
2019/MR/T018984/1. Monash University funds the
SEACO health and demographic surveillance
system. The funders of the study had no role in
study design, data collection, data analysis, data
interpretation, or writing of the report. Co-authors
of this study are also supported by the National
Institute for Health and Care Research Bristol
Biomedical Research Centre (MA). The views
expressed are those of the authors and not
necessarily those of the NIHR or the Department of
Health and Social Care.

**Competing interests:** The authors have declared
that no competing interests exist.

**Abbreviations:** SEACO, South East Asia
Community Observatory; PA, Physical activity;
LPA, light intensity physical activity; MVPA,
moderate-to-vigorous physical activity; CI,
confidence interval; WHO, World Health
Organisation; LMIC, low-to-middle income
countries; BMI, body mass index; HDSS, Health
and Demographic Surveillance System; SEACO-
CH20, SEACO Child Health update 2020; MPA,
moderate intensity physical activity; VPA, vigorous
intensity physical activity; mg, milligravities; SD,
standard deviation; MYR, Ringgit Malaysia; β, beta
coefficient; SE, standard error; Min/day, minutes
per day.

2.7–2.9) hours/day in LPA, and 33.0 (95% CI = 31.0–35.1) minutes/day in MVPA. Greater
PA and less time inactive were observed in boys vs girls, children vs adolescents, Indian
and Chinese vs Malay children and higher income vs lower income households. Data collec-
tion during Ramadan or during COVID-19 restrictions were not associated with MVPA
engagement after adjustment for demographic characteristics.

## Conclusions

Demographic characteristics remained the strongest correlates of accelerometer-measured
24-hour movement behaviours in Malaysian children and adolescents. Future studies
should seek to understand why predominantly girls, adolescents and children from Malay
ethnicities have particularly low movement behaviours within Malaysia.

## Introduction

Being physically active is associated with better physical and mental health [1, 2] and non-
communicable disease prevention in children [3–6]. The World Health Organisation (WHO)
2020 physical activity (PA) guidelines recommend that children and adolescents aged 5–17
years participate in an average of 60 minutes per day (min/day) of moderate-to-vigorous
intensity PA (MVPA) across the week [7–9]. Despite this, many children and adolescents in
both high income and low-to-middle income countries (LMICs) worldwide do not meet PA
guidelines and accrue high levels of time spent inactive [10–14]. PA has been summarised
globally, with one study finding 79–85% of a sample of 1.6 million 11–17 year olds from 146
countries participated in insufficient PA, with no clear pattern by country income [13]. How-
ever, this global summary used self-report measures of PA, which are known to exhibit bias;
therefore, there is a need for population-representative data, particularly in LMICs, using
device-based measures of PA [15].

Quantifying movement behaviours over 24-hours enables estimation of the combined
effects and inter-relations of sleep, inactive time and PA, which represent dependent inter-
changeable behaviours occurring within a finite period that are compositional in nature [16,
17]. Studies investigating 24-hour accelerometer-measured PA in children and adolescents
tend to be in high-income countries, with few studies conducted in LMICs [18]. Existing stud-
ies report mean MVPA varied from less than 40 min/day in UK adolescents [19, 20], to more
than 70 min/day in Mexican and Kenyan children and adolescents [18, 21]. While some stud-
ies have found high levels of MVPA in children from LMICs [18, 21], others in middle income
countries estimated lower MVPA engagement, e.g., 45 and 49 min/day in India and China
respectively [18]. Current data suggest that PA levels are not consistently related to country
geography, ethnicity, or income levels [14, 22], with the limited evidence across LMICs incon-
clusive, therefore individual countries should measure their own trends to target public health
resources effectively.

To our knowledge, Malaysia is a LMIC with no data on 24-hour movement behaviours.
Current knowledge of PA engagement in Malaysian children and adolescents is highly variable
[23]. Pedometer studies suggest children's step counts are below guidelines (13,000 steps for
boys and 11,000 steps for girls [24]) [25–28]. But specific estimates of MVPA range from 9–27
min/day or 60–70 min/day, based on data from accelerometers worn during waking hours
[12, 29] or self-reported surveys [30], respectively. Malaysian youth may have substantially
lower levels of PA than children and adolescents from high income and other LMICs [12],

which could be a real concern for child PA levels and health in Malaysia. However, conclusions are complicated by the use of different age ranges and different measures of PA within studies. While self-report measures can provide important contextual information about the type of PA children engage in [31], they cannot accurately or reliably quantify non-ambulatory activities, intensity or temporal PA patterns [32], meaning 24-hour movement behaviours cannot be estimated. Accelerometers can reduce bias from reporting errors and quantify intensity and temporal patterns of PA but the two existing studies, to our knowledge, that have used accelerometers in small samples of Malaysian children only captured data during waking hours. Non-wear time is inappropriately allocated as sleep in these studies [33]. Sleep, inactive time, time in light intensity PA (LPA) and MVPA are independently beneficial for different aspects of health [11, 34–36], but are not available for Malaysian children and adolescents based on research.

To fully understand public health needs and design more effective movement behaviour interventions, data captured over 24-hours is required [18, 37, 38]. Such research could help understand levels, context and patterns of PA, and how time is spent in PA across the whole 24-hour day in Malaysian children and adolescents. Furthermore, it is not fully understood how demographic characteristics (i.e., sex, age, ethnicity, body mass index [BMI] category, socioeconomic status) and external events (i.e., Ramadan, COVID-19 related restrictions) relate to 24-hour PA behaviours in Malaysian children. This may help identify specific groups at risk of low PA, and allow for the design of targeted interventions to increase PA with potentially positive effects on health [23, 39, 40]. Therefore, the primary aim of this study was to explore PA and 24-hour movement behaviours in Malaysian children and adolescents using wrist-worn accelerometers, together with self-reported PA (using the PAQ-C questionnaire). We investigated differences in time spent in sleep, inactivity, LPA, and MVPA between demographic groups and by external events.

## Materials and methods

### Study design and participants

South East Asia Community Observatory (SEACO) Health and Demographic Surveillance System (HDSS) is a dynamic cohort of ≈13,500 households in Segamat, Malaysia established in 2011 [41]. SEACO has conducted health surveys on ≈25,000 adults and children in 2013 and 2018. The SEACO Child Health update 2020 (SEACO-CH20) study aimed to collect health, PA and diet information from Malaysian children and adolescents in a subsample of the main SEACO cohort. Children aged 7–18 years that previously participated in SEACO were eligible to take part, however owing to resource and COVID-19 pandemic limitations, we aimed to recruit a random subsample of up to 1100 children stratified by age group. Safety measures were put in place due to data collection taking place during the COVID-19 pandemic to minimise risks to participants, households and data collectors, which meant that the location of the household also limited eligibility.

Fieldworkers conducted face-to-face data collection visits at the selected households. Informed consent was obtained from parents/guardians on behalf of their child and children were asked to provide assent. Ethical approval was obtained from the Monash University Human Research Ethics Committee on 17/03/2020 (Project ID: 23271).

### Measures

**Accelerometer physical activity.** Participants were asked to wear Axivity AX6 triaxial accelerometers on the "wrist they use to write" for 24 hours/day over 7 days. This placement site was chosen due to high acceptability and compliance of wrist-worn accelerometers in

children [42], and it being a commonly used placement site in large-scale population-based cohort studies [43], allowing for greater comparability of our results. Participants could remove the accelerometer for religious observations and were instructed on how to replace it afterwards. Accelerometers were set up to record data at a frequency of 100Hz, and then the collected data was converted to 5 second epochs. Accelerometer processing was conducted using the GGIR package v2.7.7 [44] in R version 4.1.2, which uses data from every available hour to make the best possible inference on average metric value per average day. Non-wear was imputed in GGIR by default using average data at similar time points during valid days [45], and sleep was defined using the default sleep algorithm. Data was classified as minutes (min/day) and average proportion of 24 hours spent in sleep, inactive time, LPA, moderate PA (MPA) and vigorous PA (VPA) (combined also to calculate MVPA) outcomes using widely-used validated cut points for children [46].

To describe the intensity profile of an individual, we calculated the intensity gradient (negative curvilinear relationship between intensity and time accumulated in that intensity which is log transformed into a linear relationship). A steeper gradient indicates more time in higher intensity activities [47]. Average acceleration (the mean of all worn and imputed acceleration values over the wear period, measured in milligravities [mg]) was also estimated [45, 48]. These non-cut point metrics capture how fast PA changes over time (intensity gradient) and the total volume of accumulated PA (average acceleration) [48, 49]. Traditional cut-point metrics enable the interpretation of PA in terms of WHO MVPA guidelines. While reporting of non-cut-point metrics (intensity gradient and average acceleration) will maximise the comparability of PA data across studies using different populations, accelerometer models and processing decisions, to allow for future data harmonisation [46, 48, 50, 51].

**Self-report physical activity.** The PAQ-C questionnaire is a 7-day recall instrument developed to assess PA in children and adolescents [52]. The Malaysian version of the PAQ-C questionnaire has shown adequate validity and good internal consistency [53–55]. It consists of nine items (scored on a 5-point Likert scale, a higher score in each item indicating a higher activity level) asking about PA at different times, and during specific activities [52]. Responses were averaged to calculate a total PAQ-C activity score and scores for PAQ-C domains (calculated using grouped scores of relevant items): Organised/structured PA, Physical Education related PA, School recreational PA, Outside school PA and Weekend PA [56]. Although PAQ-C domains are scored on a 1–5 scale, scales are different for each item, so scores are not clinically meaningful, however can be interpreted as a higher score = more PA of that domain. Participants were asked to complete this questionnaire at any point during the study week, so PAQ-C measures represent 'habitual' PA.

**Demographic characteristics.** Child age, gender and ethnicity were extracted from the SEACO 2018 health survey. Children were grouped into two age groups: childhood (7–12 years) and adolescence (13–18 years), corresponding to Malaysian primary and secondary school ages [57]. Ethnicity was reported as Malay, Chinese, Indian, or other. Data collectors measured height and weight using a Transtek digital weighing scale and height gauge and BMI was calculated and converted to age-adjusted standardised z-scores using the WHO 2007 BMI reference for children aged 5–19 (BMI z-score) [58]. Children were classified as underweight, overweight and obese according to WHO definitions [58, 59], if the standardised BMI z-score was <-2, >1 and < = 2, and >2 standard deviations (SD) from the mean, respectively, with remaining children classified as a healthy weight.

**Change in BMI and obesity incidence.** Participant BMI was also measured in a previous SEACO Health Round survey in 2018. The 2018 and 2022 BMI information were used to calculate the change in BMI z-score and four-year average obesity incidence (percentage of participants with obesity in 2022 out of those without obesity in 2018).

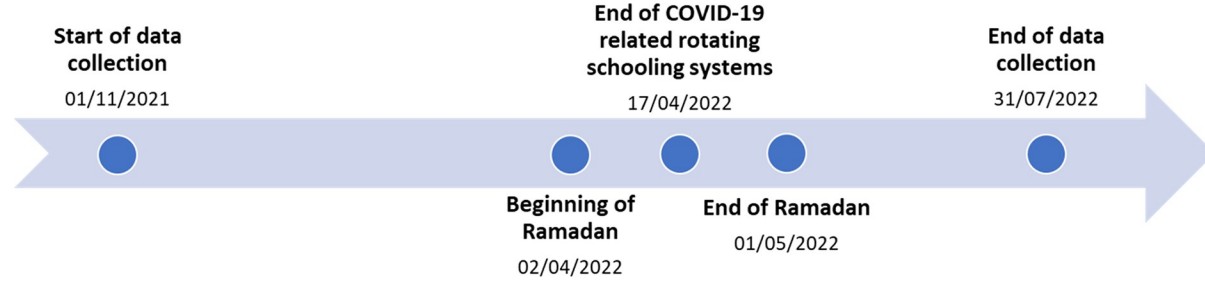

**Fig 1. Timeline of data collection including external events.**

**Household socioeconomic variables.** Parent-reported socioeconomic characteristics (highest education level in household, monthly household income) were extracted from the SEACO 2018 health survey. The highest education level in household was grouped as "up to secondary" and "tertiary" (i.e., degree level or higher) education. Monthly household income was grouped as Ringgit Malaysia (MYR)<2000 and MYR≥2000 for analysis, an income threshold equivalent to the lowest 10% household income decile (B1) according to the latest Johor, Malaysia Household Income Survey [60].

**External events.** Fig 1 shows the timeline of data collection in relation to external events that may have impacted PA. External events included participants with data collection: 1) during Ramadan (3rd April– 1st May 2022); 2) whilst COVID-19-related rotation schooling systems were in place in Malaysia (beginning of data collection - 17th April 2022) [61]. COVID-19-related rotation schooling systems involved students in each class being split into 2 groups, with each group taking turns to attend school physically on a weekly basis (i.e., children physically attend school on alternate weeks). Participants were grouped using the date of accelerometer wear, whereby only participants with entire data collection during Ramadan/COVID-19-related rotation schooling systems included in these subgroups.

## Statistical analysis

Descriptive statistics (categorical variables: percentages; continuous variables: means and SD for normally distributed variables and median and inter-quartile range otherwise), were calculated for demographic variables. Missing data are reported in S1 Table. Means, SD and 95% confidence intervals (95% CI) were calculated for daily time spent in MVPA, LPA, inactive time and sleep as well as intensity gradient, average accelerations, PAQ-C score and PAQ-C domains.

We repeated descriptive analysis and investigated differences by comparing means and 95% CI in accelerometer and self-report PA by demographic characteristics (sex, age groups, ethnicities, BMI groups, socioeconomic groups [highest education level in household and monthly household income]), and external events/factors (Ramadan data collection and COVID-19 related rotation schooling systems). To confirm if PA behaviours differed by demographics, we conducted univariate linear regression models with the PA measures (MVPA/LPA/inactive time/sleep) as the dependent variables and the demographic variable as the independent variable. Sensitivity analysis excluding data collected during Ramadan was conducted for ethnicity regression models, as those of Malay ethnicity were more likely to be fasting during Ramadan.

To confirm if PA behaviours differed by external events/factors, we compared multiple linear regression models unadjusted (Model 1: dependent variable = MVPA (min/day) and independent variable = External event) and adjusted (Model 2a: model 1 plus confounders = age

and sex, Model 2b: model 1 plus confounders = age, sex and ethnicity) for demographics Due to the association between ethnicity and observing Ramadan, ethnicity was not included in models investigating associations with Ramadan. Assumptions were checked by residual plots. Analyses were undertaken using Stata v17 [62].

## Results

Fig 2 illustrates the flow of participants from recruitment to data analysis. We approached 1993 households, of which 728 people consented to participate, 626 participants were included overall and 608 wore accelerometers. A total of 19% had insufficient valid accelerometer data

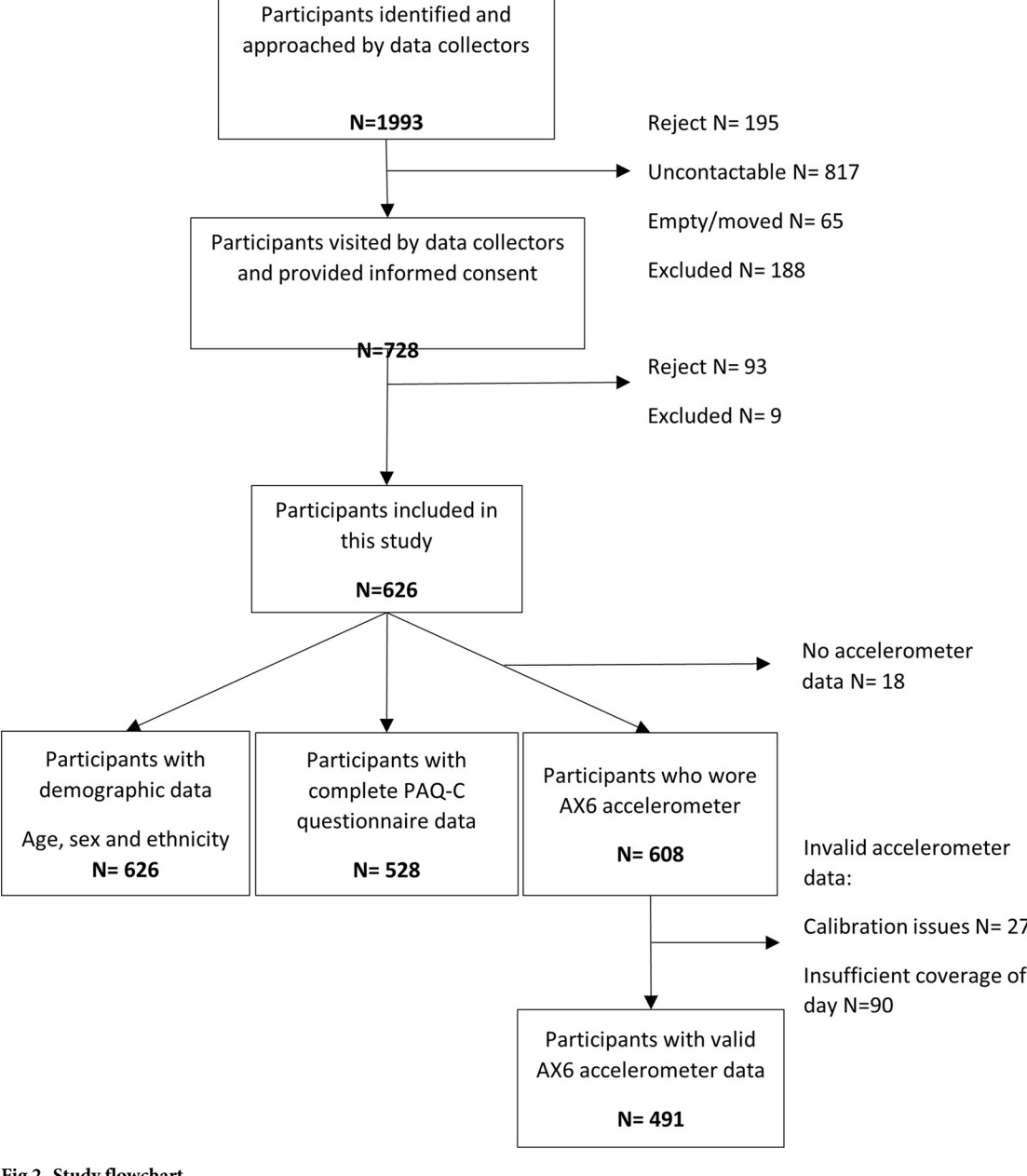

**Fig 2. Study flowchart.**

to be included in analysis due to: non-wear (8%), calibration problems (4%) and forgetting to put the monitor back on (6%) and so were excluded, leaving an accelerometer analysis sample of n = 491. Missing data varied from 0–1.6% for demographics and from 0.6–16% for PA variables (S1 Table).

Table 1 displays demographic characteristics of the children and adolescents in SEA-CO-CH20. A total of 44.1% (n = 276) participants were aged 8–12 years (children) and 55.9% (n = 350) aged 13–18 years (adolescents), with 49% girls. Of these, 67% of participants were Malay, 21% Chinese and 13% Indian, which is comparable to the general Segamat population [41, 63]. A total of 4% of participants were underweight, 56% healthy weight, 17% overweight and 23% had obesity, with an average BMI z-score of 0.61. Although overweight or obesity prevalence was stable between 2018 and 2021/2022 data collection (≈40%), there was an increase in mean BMI z-score and annual obesity incidence of 2.25% between 2018 and 2021/2022. Demographic characteristics were similar for participants included in the final sample (n = 491) compared with both the total sample (n = 626) and the eligible sample (n = 1993), apart from a higher proportion of Malay ethnicity (70% in total sample) and fewer Chinese ethnicity (18% in total sample) participants in the final sample compared with the total sample.

## Accelerometer-measured physical activity

The study population spent 489.8 min/day (≈8.2 hours/day) (95% CI = 476.7–502.8 min/day) asleep, 746.8 min /day (≈12.4 hours/day) (95% CI = 733.9–759.6 min/day) inactive, 170.1 min /day (≈2.8 hours/day) (95% CI = 164.6–175.7 min/day) in LPA, and 33.0 min/day (95% CI = 31.0–35.1 min/day) in MVPA (Table 2), equating to 34%, 52%, 12% and 2.3% of the day spent asleep, inactive, in LPA and in MVPA, respectively (Fig 3 and S2 Table). In total, only 13% of the study population met WHO MVPA guidelines (S3 Table). The mean accelerometer-measured PA intensity gradient was -2.27 (95% CI = -2.29 to -2.25), and the average acceleration was 39.2mg (95% CI = 37.62–40.79mg) (S4 Table).

We observed more MVPA among boys vs girls ($\beta$ = 12.8 [95% CI = 8.7–16.8] min/day), children vs adolescents ($\beta$ = 20.2 [95% CI = 16.4–24.0] min/day), Indian vs Malay ethnic groups ($\beta$ = 12.6 [95% CI = 6.3–18.9] min/day), Chinese vs Malay ethnic groups ($\beta$ = 6.2 [95% CI = 0.7–11.6] min/day), and higher income (≥2000MYR/month) vs lower income (<2000MYR/month) households ($\beta$ = 4.3 [95% CI = 0.02–8.5] min/day) (Table 2 and S5 Table and Fig 4). Although when only including participants with data collection outside of Ramadan, differences in MVPA between Malay vs Chinese ethnicity participants were attenuated ($\beta$ = 4.5 [95% CI = -1.3–10.2] min/day) (S6 Table). Groups defined by BMI status and household education level were similar (Table 2 and S5 Table and Fig 4). LPA was greater among children vs adolescents ($\beta$ = 37.4 [95% CI = 26.8–48.0] min/day) and Indian vs Malay ($\beta$ = 41.6 [95% CI = 25.1–58.2] min/day) ethnic groups but no difference by sex, BMI groups, or household socioeconomic position. Inactive time was greater in females vs males ($\beta$ = 31.3 [95% CI = 5.7–56.8] min/day), adolescents vs children ($\beta$ = 59.9 [95% CI = 34.7–85.2] min/day) and Malay vs Indian ($\beta$ = 72.6 [95% CI = 33.9–111.4] min/day) groups only. Finally, we observed no notable differences in sleep duration between demographic groups (Table 2 and S5 Table). Residual plots showed no evidence of violation of model assumptions. Regarding non-cut-point PA metrics (S4 Table), we observed more favourable intensity gradients and higher average acceleration in boys vs girls, children vs adolescents and in participants from higher (MYR≥2000) vs lower (MYR<2000) income households. While Indians had greater average acceleration than Malay participants (by 11mg), intensity gradients were similar.

**Table 1. Sample demographic characteristics.**

|  |  | Descriptive statistic (mean, SD or %) | Number of participants | N = 491 with useable accelerometer data |
|---|---|---|---|---|
| **Age (mean, SD)** |  | 13.0 (2.9) | 626 | 12.9 (2.9) |
| **Age (% child <13years)** |  | 44.1 | 626 | 45.6 |
| **Sex (% Female)** |  | 48.7 | 626 | 50.7 |
| **Ethnicity (%):** |  |  |  |  |
|  | Malay | 66.6 | 417 / 626 | 69.5 |
|  | Chinese | 20.6 | 129 / 626 | 17.9 |
|  | Indian | 12.6 | 79 / 626 | 12.6 |
|  | Other | 0.2 | 1 / 626 | 0 |
| **Height (cm) (mean, SD)** |  | 149.2 (13.8) | 625 | 148.5 (13.6) |
| **Weight (kg) (mean, SD)** |  | 48.8 (18.2) | 625 | 48.0 (17.5) |
| **BMI z-score 2022 (mean, SD)** |  | 0.61 (1.58) | 625 | 0.61 (1.54) |
| **BMI category 2022 (%):** |  |  |  |  |
|  | Underweight | 3.8 | 24 | 3.5 |
|  | Healthy Weight | 55.8 | 349 | 56.3 |
|  | Overweight | 17.1 | 107 | 18.0 |
|  | Obese | 23.2 | 145 | 22.2 |
| **BMI z-score 2018 (mean, SD)** |  | 0.59 (1.50) | 622 | 0.54 (1.46) |
| **BMI category 2018 (%):** |  |  |  |  |
|  | Underweight | 2.9 | 18 | 3.1 |
|  | Healthy Weight | 57.7 | 359 | 59.2 |
|  | Overweight | 19.5 | 121 | 19.7 |
|  | Obese | 19.9 | 124 | 18.0 |
| **Change in BMI z-score (mean, SD)** |  | 0.02 (0.96) | 621 | 0.07 (0.95) |
| **Annual obesity incidence (%)** |  | 2.25 | 621 | 2.18 |
| **Highest education level in household (%):** |  |  | 624 |  |
|  | Up to Primary | 3.9 | 24 | 3.7 |
|  | Secondary | 69.6 | 434 | 70.0 |
|  | Tertiary | 26.6 | 166 | 26.3 |
| **Monthly household income 2018 (MYR) (mean, SD)** |  | 2398.5 (1696.4) | 616 | 2406.4 (1736.5) |
| **Accelerometer PA (mean, SD):** |  |  |  |  |
|  | Sleep (mins/day) | 489.8 (147.4) | 491 | 489.8 (147.4) |
|  | Inactive time (mins/day) | 746.8 (144.7) | 491 | 746.8 (144.7) |
|  | LPA (mins/day) | 170.1 (62.3) | 491 | 170.1 (62.3) |
|  | MPA (mins/day) | 30.4 (20.7) | 491 | 30.4 (20.7) |
|  | VPA (mins/day) | 2.7 (3.7) | 491 | 2.7 (3.7) |
|  | MVPA (mins/day) | 33.0 (23.5) | 491 | 33.0 (23.5) |
|  | Intensity Gradient | -2.27 (0.23) | 490 | -2.27 (0.23) |
|  | Average Acceleration (mg) | 39.2 (17.9) | 491 | 39.2 (17.9) |
| **PAQ-C Score (1–5) (mean, SD)** |  | 2.2 (0.7) | 528 | 2.2 (0.7) |
| **PAQ-C Domains (1–5) (mean, SD):** |  |  |  |  |
|  | Organised/structured PA | 1.3 (0.3) | 591 | 1.4 (0.3) |
|  | Physical Education related PA | 2.4 (1.3) | 622 | 2.3 (1.3) |
|  | School recreational PA | 1.8 (0.9) | 600 | 1.8 (0.9) |
|  | Outside school PA | 2.6 (1.2) | 603 | 2.7 (1.2) |

*(Continued)*

**Table 1.** (Continued)

| | Descriptive statistic (mean, SD or %) | Number of participants | N = 491 with useable accelerometer data |
|---|---|---|---|
| *Weekend PA* | 2.7 (1.2) | 610 | 2.7 (1.1) |

Note: BMI = Body Mass Index, SD = standard deviation, PA = physical activity, LPA = light intensity physical activity, MPA = moderate physical activity, VPA = vigorous physical activity, MVPA = moderate to vigorous intensity physical activity, MYR = Ringgit Malaysia. The PAQ-C questionnaire is scored on a 5-point Likert scale, with a higher score indicating higher level of activity.

## Self-report physical activity

Table 3 summarises self-reported physical activity. Findings were broadly similar to accelerometer-measured PA, with boys reporting greater participation in organised/structured PA, school recreational PA, outside school PA and weekend PA than girls, and children self-reporting more PA than adolescents across all PAQ-C domains. As with accelerometer-measured PA, Indian participants self-reported more PA participation, overall and within most domains. However, not captured by accelerometers, we also observed that Chinese participants reported significantly less outside school and weekend PA and greater engagement in

**Table 2. Accelerometer-measured physical activity (minutes/day) split by demographic characteristics.**

| | Number of participants | Sleep duration (min/day) | | Inactive time (min/day) | | LPA (min/day) | | MVPA (min/day) | | MPA (min/day) | | VPA (min/day) | |
|---|---|---|---|---|---|---|---|---|---|---|---|---|---|
| | | mean | 95% CI | mean | 95% CI | mean | 95% CI | mean | 95% CI | mean | 95% CI | mean | 95% CI |
| *Total* | 491 | 489.8 | 476.7–502.8 | 746.8 | 733.9–759.6 | 170.1 | 164.6–175.7 | 33.0 | 31.0–35.1 | 30.4 | 28.5–32.2 | 2.7 | 2.4–3.0 |
| *Sex* | | | | | | | | | | | | | |
| *Male* | 242 | 497.8 | 478.8–516.8 | 730.9 | 712.1–749.7 | 171.5 | 163.6–179.4 | 39.5 | 36.4–42.7 | 35.8 | 33.1–38.6 | 3.7 | 3.1–4.2 |
| *Female* | 249 | 482.0 | 463.9–500.0 | 762.2 | 744.8–779.6 | 168.8 | 161.0–176.5 | 26.8 | 24.2–29.3 | 25.1 | 22.8–27.3 | 1.7 | 1.4–2.0 |
| *Age* | | | | | | | | | | | | | |
| *Child* | 224 | 491.0 | 475.0–507.1 | 714.2 | 698.3–730.1 | 190.4 | 183.5–197.4 | 44.0 | 41.0–47.1 | 40.0 | 37.3–42.7 | 4.0 | 3.5–4.6 |
| *Adolescent* | 267 | 488.7 | 468.7–508.7 | 774.1 | 755.2–793.0 | 153.1 | 145.3–160.9 | 23.8 | 21.5–26.1 | 22.2 | 20.2–24.3 | 1.6 | 1.2–1.9 |
| *Ethnicity* | | | | | | | | | | | | | |
| *Malay* | 341 | 491.4 | 476.2–506.5 | 754.5 | 739.6–769.4 | 163.3 | 156.7–169.9 | 30.4 | 28.0–32.7 | 28.0 | 25.9–30.1 | 2.3 | 2.0–2.7 |
| *Chinese* | 88 | 469.1 | 441.9–496.2 | 762.4 | 734.8–790.0 | 172.0 | 160.3–183.7 | 36.5 | 31.4–41.7 | 33.1 | 28.7–37.4 | 3.5 | 2.5–4.4 |
| *Indian* | 62 | 510.2 | 461.1–559.4 | 681.9 | 637.7–726.1 | 204.9 | 189.3–220.6 | 42.9 | 36.2–49.7 | 39.5 | 33.5–45.5 | 3.4 | 2.5–4.4 |
| *BMI Category* | | | | | | | | | | | | | |
| *Underweight* | 17 | 512.7 | 471.4–553.9 | 730.1 | 667.2–792.9 | 161.7 | 128.2–195.1 | 35.6 | 21.3–50.0 | 32.8 | 20.2–45.4 | 2.8 | 0.9–4.7 |
| *Healthy weight* | 276 | 492.1 | 475.1–509.1 | 749.7 | 732.9–766.6 | 167.1 | 159.4–174.9 | 30.7 | 27.9–33.5 | 28.1 | 25.7–30.5 | 2.6 | 2.1–3.1 |
| *Overweight* | 88 | 483.3 | 452.7–514.0 | 742.0 | 711.5–772.5 | 178.3 | 165.8–190.9 | 36.1 | 31.4–40.8 | 33.2 | 29.0–37.4 | 2.9 | 2.2–3.6 |
| *Obese* | 109 | 486.1 | 454.3–518.0 | 746.1 | 716.3–775.8 | 171.7 | 161.1–182.3 | 35.8 | 31.5–40.1 | 33.1 | 29.2–36.9 | 2.7 | 2.1–3.3 |
| *Highest education level in household* | | | | | | | | | | | | | |
| *Up to Secondary* | 361 | 485.8 | 470.3–501.4 | 752.7 | 737.5–767.8 | 169.1 | 162.3–175.8 | 32.1 | 29.7–34.6 | 29.6 | 27.5–31.8 | 2.5 | 2.1–2.9 |
| *Tertiary* | 129 | 501.6 | 477.3–525.8 | 728.7 | 704.5–752.9 | 173.7 | 164.4–183.1 | 35.7 | 31.5–39.9 | 32.6 | 28.9–36.2 | 3.2 | 2.5–3.8 |
| *Monthly household income 2018 (MYR)* | | | | | | | | | | | | | |
| *<2000* | 270 | 483.8 | 464.8–502.8 | 755.9 | 737.8–773.9 | 168.7 | 161.0–176.3 | 31.2 | 28.4–34.1 | 28.9 | 26.4–31.3 | 2.4 | 2.0–2.8 |
| *≥2000* | 216 | 498.1 | 480.2–516.0 | 734.4 | 716.0–752.8 | 171.8 | 163.7–180.0 | 35.5 | 32.4–38.6 | 32.4 | 29.7–35.2 | 3.1 | 2.6–3.6 |

Note: LPA = light intensity physical activity, MVPA = moderate to vigorous intensity physical activity, MPA = moderate physical activity, VPA = vigorous physical activity, BMI = body mass index, CI = confidence interval, MYR = Ringgit Malaysia

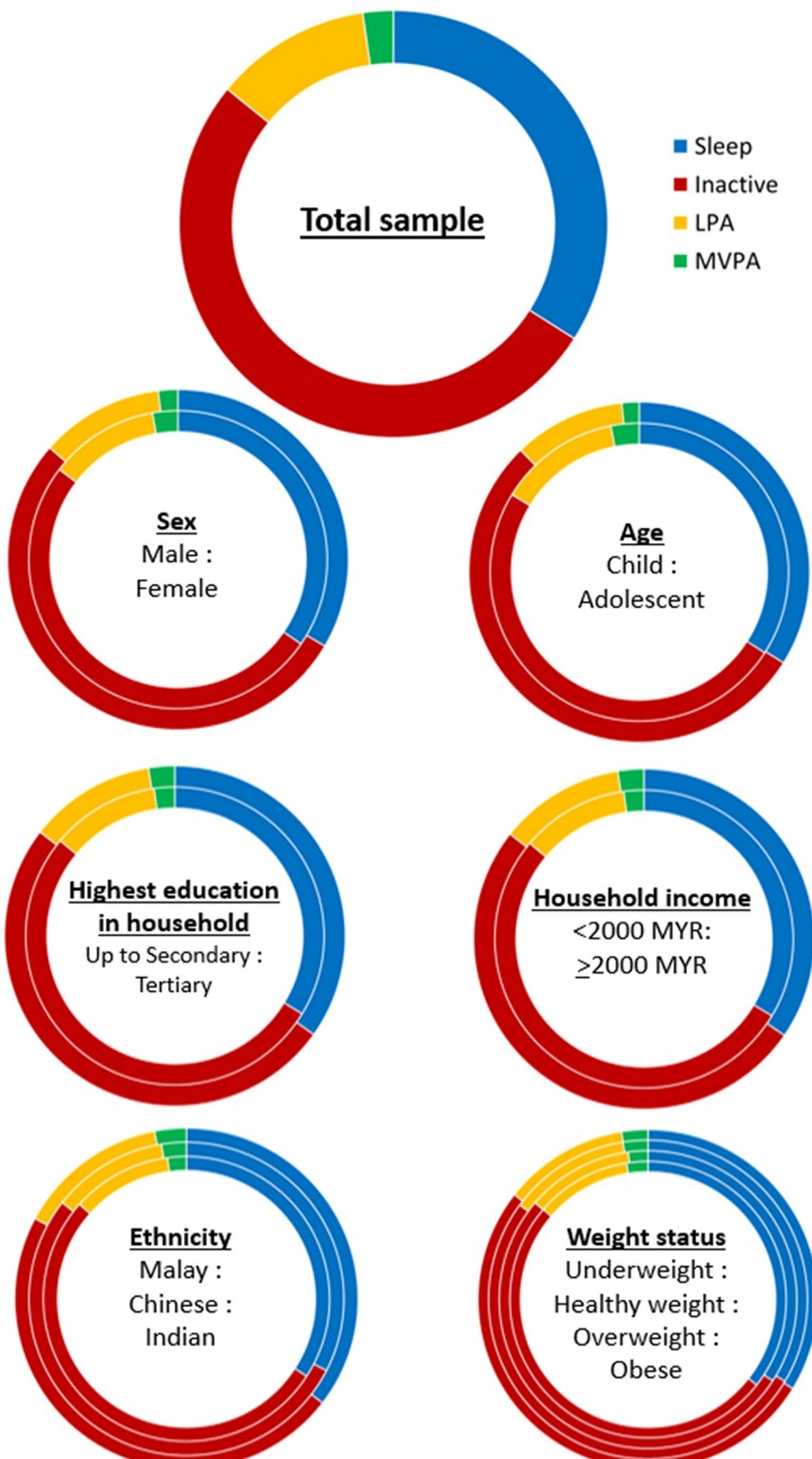

**Fig 3. Proportions of time (% of day) spent in different movement behaviours by demographic characteristic group.**

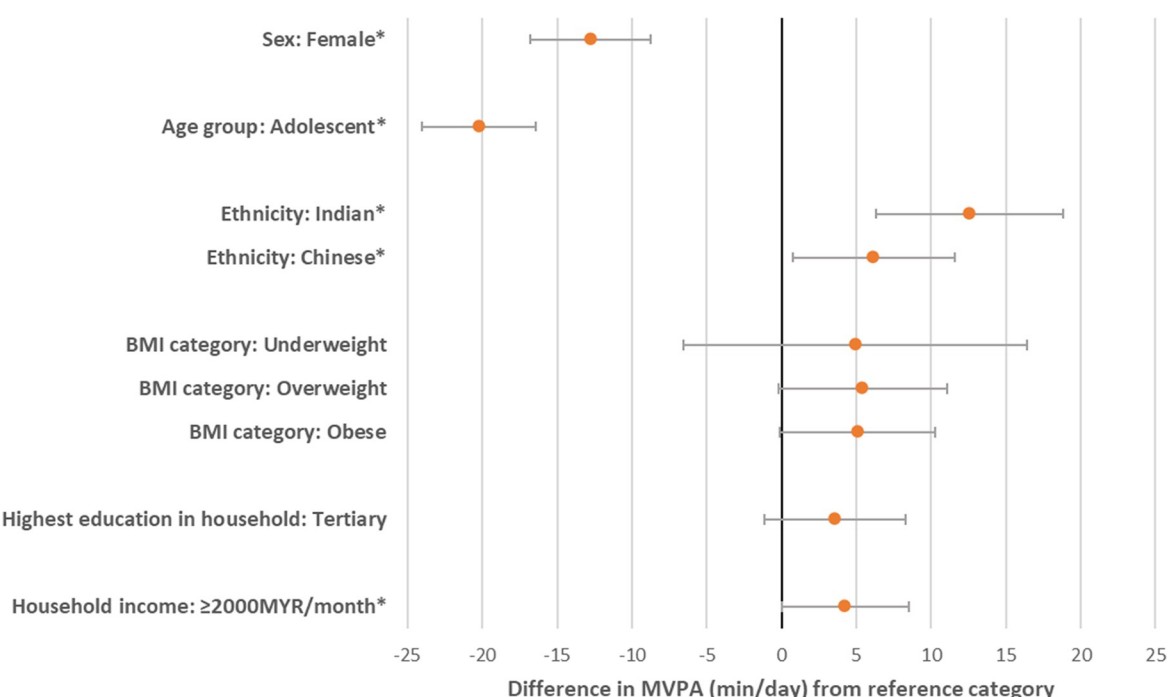

**Fig 4. Univariate linear regression results of differences in MVPA engagement from reference category.**

physical education-related PA compared non-Chinese groups. Finally, children with higher household income reported greater physical education-related PA, although PAQ-C score was similar to children from lower-income households.

## Variation in physical activity by external events

Participants with data collection during Ramadan (n = 62) and during COVID-19-related rotating school systems (n = 254) engaged in less accelerometer-measured MVPA and LPA and had lower PAQ-C scores, especially for the physical education related PAQ-C domain (S7 and S8 Tables). Unadjusted linear models estimated MVPA was 6.9 (95% CI = 0.6–13.2) min/day lower for those with data collection during Ramadan, and 6.6 (95% CI = 2.3–10.9) min/day lower for those with data collection during COVID-19-related rotating school systems. However, when COVID-19-related rotating school systems models were adjusted for age, sex and ethnicity (Model 2b), evidence of associations were attenuated (β = 3.5 [95% CI = -1.8–8.8] min/day). When Ramadan models were adjusted for age and sex (Model 2a), associations with MVPA were also no longer significant (β = 5.1 [95% CI = -0.4–10.6] min/day), suggesting that the demographic characteristics of participants in data collection during external events were more strongly associated with their PA (S9 Table).

## Discussion

To our knowledge, this is the first study investigating 24-hour movement behaviours measured continuously over 7 days among both children and adolescents in South East Asia. Our findings suggest Malaysian children and adolescents engage in relatively low levels of MVPA. We observed higher MVPA among children vs adolescents, males vs females, Indian and Chinese vs Malay ethnic groups and children from higher income vs lower income households. Our data collection spanned a dynamic period with external events that may affect PA, such as

**Table 3. PAQ-C measured physical activity split by demographic characteristics.**

| | PAQ-C score | | Organised/ structured PA | | Physical Education related PA | | School recreational PA | | Outside school PA | | Weekend PA | |
|---|---|---|---|---|---|---|---|---|---|---|---|---|
| | mean | 95% CI | mean | 95% CI | mean | 95% CI | mean | 95% CI | mean | 95% CI | mean | 95% CI |
| *Total* | 2.20 | 2.14–2.26 | 1.35 | 1.33–1.37 | 2.37 | 2.27–2.47 | 1.82 | 1.75–1.89 | 2.62 | 2.53–2.71 | 2.70 | 2.61–2.79 |
| *Sex* | | | | | | | | | | | | |
| *Male* | 2.35 | 2.26–2.43 | 1.40 | 1.36–1.43 | 2.43 | 2.29–2.58 | 2.01 | 1.90–2.13 | 2.80 | 2.67–2.93 | 2.85 | 2.72–2.98 |
| *Female* | 2.05 | 1.97–2.12 | 1.30 | 1.27–1.32 | 2.30 | 2.15–2.45 | 1.61 | 1.53–1.70 | 2.43 | 2.30–2.56 | 2.54 | 2.41–2.67 |
| *Age* | | | | | | | | | | | | |
| *Child* | 2.41 | 2.32–2.50 | 1.42 | 1.39–1.45 | 2.65 | 2.49–2.81 | 2.00 | 1.88–2.12 | 2.91 | 2.76–3.05 | 2.87 | 2.73–3.02 |
| *Adolescent* | 2.05 | 1.98–2.12 | 1.29 | 1.26–1.32 | 2.15 | 2.01–2.29 | 1.68 | 1.59–1.76 | 2.40 | 2.29–2.52 | 2.56 | 2.44–2.68 |
| *Ethnicity* | | | | | | | | | | | | |
| *Malay* | 2.19 | 2.12–2.27 | 1.34 | 1.31–1.36 | 2.01 | 1.90–2.13 | 1.81 | 1.72–1.90 | 2.71 | 2.60–2.83 | 2.73 | 2.62–2.84 |
| *Chinese* | 2.03 | 1.92–2.14 | 1.28 | 1.25–1.31 | 3.17 | 2.94–3.40 | 1.72 | 1.60–1.84 | 2.02 | 1.85–2.19 | 2.37 | 2.18–2.57 |
| *Indian* | 2.55 | 2.38–2.71 | 1.54 | 1.47–1.61 | 2.95 | 2.66–3.24 | 2.03 | 1.79–2.26 | 3.11 | 2.88–3.34 | 3.03 | 2.76–3.29 |
| *BMI Category* | | | | | | | | | | | | |
| *Underweight* | 2.02 | 1.72–2.31 | 1.30 | 1.20–1.39 | 2.42 | 1.85–2.99 | 1.60 | 1.33–1.88 | 2.30 | 1.81–2.80 | 2.63 | 2.15–3.10 |
| *Healthy weight* | 2.19 | 2.11–2.27 | 1.34 | 1.31–1.37 | 2.32 | 2.19–2.46 | 1.83 | 1.73–1.93 | 2.62 | 2.49–2.74 | 2.68 | 2.55–2.81 |
| *Overweight* | 2.21 | 2.07–2.35 | 1.35 | 1.30–1.40 | 2.41 | 2.15–2.68 | 1.84 | 1.67–2.02 | 2.61 | 2.39–2.82 | 2.79 | 2.58–3.01 |
| *Obese* | 2.25 | 2.12–2.37 | 1.38 | 1.33–1.42 | 2.45 | 2.23–2.67 | 1.82 | 1.67–1.98 | 2.67 | 2.47–2.88 | 2.66 | 2.48–2.85 |
| *Highest education level in household* | | | | | | | | | | | | |
| *Up to Secondary* | 2.22 | 2.15–2.29 | 1.35 | 1.33–1.38 | 2.32 | 2.20–2.45 | 1.80 | 1.71–1.88 | 2.63 | 2.52–2.74 | 2.74 | 2.63–2.84 |
| *Tertiary* | 2.15 | 2.04–2.26 | 1.35 | 1.31–1.38 | 2.49 | 2.29–2.70 | 1.88 | 1.74–2.03 | 2.61 | 2.43–2.79 | 2.61 | 2.43–2.79 |
| *Monthly household income 2018 (MYR)* | | | | | | | | | | | | |
| *<2000* | 2.17 | 2.09–2.25 | 1.34 | 1.31–1.37 | 2.20 | 2.06–2.33 | 1.84 | 1.73–1.94 | 2.56 | 2.43–2.68 | 2.63 | 2.50–2.76 |
| *≥2000* | 2.25 | 2.16–2.33 | 1.35 | 1.32–1.38 | 2.56 | 2.40–2.73 | 1.81 | 1.71–1.91 | 2.72 | 2.58–2.85 | 2.79 | 2.66–2.93 |

Note: PA = physical activity, BMI = body mass index, CI = confidence interval, MYR = Ringgit Malaysia. The PAQ-C questionnaire is scored on a 5-point Likert scale, with a higher score indicating higher level of activity.

Ramadan and COVID-19 movement restrictions. Although these events seemed to be related to PA engagement, demographic characteristics were the factors most associated with 24-hour movement behaviours in Malaysian children and adolescents.

## Accelerometer-measured and self-report physical activity

We found Malaysian children and adolescents had low MVPA engagement (33.0 min/day), with 87% of participants not meeting the WHO 2020 MVPA recommendations. Our MVPA findings are consistent with the IPEN Adolescent study that included Malaysian adolescents but used hip-worn ActiGraph accelerometers during waking hours only [12]. IPEN found Malaysian adolescents were the least active compared to other countries, with MVPA of 26.6 min/day [12]. While MVPA in IPEN may have been underestimated because there was no capture of PA outside "waking hours", in combination with our results, it suggests that MVPA in Malaysian adolescents is well below the recommended 60 min/day. Other studies using 24-hour wrist-worn accelerometers in children from European high-income countries observed 35–63 min/day MVPA, higher than our observations [19, 38]. Therefore, our findings suggest Malaysian children and adolescents engage in relatively low levels of MVPA compared to studies using similar protocols in children from high-income countries.

Our finding that participants spent over 12 hours/day inactive is substantially higher than that reported in the IPEN Adolescent study (9.4 hours/day inactive in Malaysian youth) [12]. Disparities may relate to the 24-hour measurement in our study, which enables the capture of inactive activities during the night-time, or due to differences in population or cut-points to determine inactive time. In comparison, another study using more comparable data collection methods, i.e., 24-hour accelerometer measurement, found similar estimates of inactive time (12.7 hours/day) in Welsh children and adolescents post-COVID-19 lockdown (April-May 2021), suggesting children's activity may have been influenced by longer-term effects of the COVID-19 pandemic similarly in Wales and Malaysia [19]. Post-pandemic recommendations and interventions are therefore equally needed across both high-income countries and LMICs to reduce inactive time among children and adolescents.

New 24-hour movement guidelines have been developed in some countries as research has shifted to focus on movement behaviours as part of a 24-hour continuum [18, 37]. Canadian guidelines include screen time (2 hour/day maximum), sleep (8–11 hours/night) and MVPA (60 min/day) recommendations [37]. Our estimates of sleep duration were within these recommendations and consistent with previous self-report findings in Malaysian children (8.4 hours/day) [26], and accelerometer-measured findings from worldwide youth (7.9 hours/day [64], 8.8 hours/day [18]). Nevertheless, we observed higher levels of variability in sleep estimates (SD = 2.45 hours/day), compared to other studies (ISCOLE study: SD = 0.9 hours/day, Malaysian study: SD = 0.83 hours/day) [18, 26]. The higher variability observed may relate to the wider age range included in our study (7–18 years). However, we recommend more studies using 24-hour accelerometry in children include sleep duration as an outcome measure [19–21], to enable further between-study comparisons.

The intensity gradient and average acceleration metrics we report are relatively novel, and as few childhood studies have used this metric, few comparisons are available. Compared to UK reference values in children aged 5–15, both the intensity gradient and average acceleration were in the lowest 10th percentile for both boys and girls at all ages [47]. This indicates that not only is the volume of activity lower in Malaysia but also the intensity is low, suggesting both volume and intensity should be targeted in future interventions to meet PA guidelines. Nevertheless, as average acceleration and intensity gradient are independently associated with health outcomes in different populations [48, 65], and as use of these metrics increases, there is a need to generate comparative data using wrist-worn accelerometers in children and adolescents from Malaysia and other LMICs [47]. This may enable higher quality PA surveillance and guidance by using metrics which allow for standardised comparisons across different study groups [47].

## Variation in physical activity by participant demographic characteristics

Consistent with previous worldwide accelerometer studies and Malaysia-based studies, we observed that boys engaged in more MVPA than girls (40 vs 27 min/day) [5, 28, 66–70]. We observed no notable difference in sleep and LPA duration between sexes, which suggests that higher MVPA displaces inactive time in boys. Findings were consistent for non-cut-point metrics and self-reported PA, with sex differences most marked for school recreational PA and outside school PA. Together, this reinforces and adds context to WHO 2020 PA guidelines for children which recommend of 60 min/day of MVPA, VPA three times per week, as well as limiting sedentary time [8, 9]. Therefore, our findings, in line with WHO guidelines, propose that intervention programs should focus on reducing inactive time and increasing PA of a higher intensity, targeting recreational activities within and outside school particularly in girls in Malaysia.

Adolescents (13–18 years) reported reduced PA across all PAQ-C domains and engaged in less accelerometer-measured PA and more inactive behaviours, compared to children (7–12 years). Most strikingly, children participated in almost twice as much MVPA as adolescents (44 vs 24 min/day) and had a steeper intensity gradient. Sleep duration was similar between groups, and so activity behaviours significantly differed during waking hours. Previous large-scale epidemiological studies and Malaysia-specific systematic reviews have reported age-related declines in PA engagement [38, 66, 68, 70, 71]. In addition, a UK study using a similar accelerometer protocol observed upper primary school children (aged 8–11 years) spent 63 min/day in MVPA, whilst those in upper secondary school (aged 14–18 years) spent only 35 min/day in MVPA [19]. The differences between child and adolescent waking hour PA behaviours in our study and others indicates the need for greater PA support and specific PA recommendations for adolescents, independent from children.

Children from different ethnicities had different PA profiles, with Indian participants spending more LPA and MVPA and less time inactive compared to their Malay counterparts. Chinese participants also spent more time in MVPA than Malay children, however these differences were attenuated in those with data collection outside of Ramadan. Non-cut-point accelerometer metrics suggested that Indian children and adolescents do not necessarily participate in more intense activities (intensity gradients were comparable with Malays), but have greater overall PA volume (average accelerations were higher than Malays). Self-report results suggested that variations in PA behaviours may be explained by increased participation in organised/structured PA (PAQ-C organised/structured PA: Indian = 1.54, Malay = 1.34), physical education PA (PAQ-C physical education-related PA: Indian = 2.95, Malay = 2.01) and outside school PA (PAQ-C outside school PA: Indian = 3.11, Malay = 2.71) by Indian children. These ethnic differences were consistent with previous findings in a study of adolescents in Peninsular Malaysia [69, 70]. However, further studies are required using larger samples of non-Malay participants, and investigating potential moderators that may impact PA participation in children of different ethnicities.

We observed few notable differences in accelerometer-measured PA between participants from different BMI categories, although a previous study found that Malaysian children with obesity engaged in less MVPA than their peers of healthy weight [29]. A recent systematic review found no association between Malaysian adolescent PA with parent education level or household income [70], although we did observe 4.3 (95% CI = 0.02–8.5) min/day greater MVPA engagement in children from higher vs lower income households. The few between-group differences in PA engagement for these characteristics in this study may be due to no real-life relationships or small subgroup sample sizes, causing a lack of statistical power to detect differences.

## Variation in physical activity by external events

We explored the relative contribution of Ramadan and COVID-19 restrictions on PA behaviours. Our results showed after adjustment for demographics, associations between Ramadan data collection and COVID-19-related rotation schooling systems with accelerometer-measured MVPA did not remain. This is in contrast to previous findings in Malaysian adults [72], and children in other countries which found pandemic effects on PA engagement [19, 73, 74]. Other studies have shown that adults who participate in Ramadan fasting engage in less PA during Ramadan [75, 76]. Together, these novel results indicate that age, sex and ethnicity remain the strongest predictors of PA engagement in Malaysian youth. External events may have a relatively negligible effect on PA participation, however, longitudinal studies where repeated measurements of PA are made in the same children so that individual characteristics are held constant, are needed to confirm our findings.

## Strengths and limitations

To our knowledge, ours is the largest study of Malaysian children and adolescents exploring accelerometer-measured PA, and the largest assessing 24-hour movement behaviours in children using wrist-worn accelerometers in an LMIC worldwide. In addition, our sample covered a broad age range and was representative of the childhood population in a typical district in Malaysia terms of ethnicity, age, and sex breakdown. The 24-hour accelerometer wear enabled interpretation of PA engagement over a 24-hour continuum for the whole week (school days and weekend days); and enables more precise assessment of sleep duration, as when accelerometers are only worn during "waking hours" without the addition of a logbook, sleep duration estimates can result in activity behaviour misclassification [77, 78]. Lastly, our use of the open-source GGIR package allows for the processing and analysis of raw accelerometer data and facilitates greater reproducibility of analysis. GGIR is not accelerometer brand specific, and its capabilities allow for a greater level of data harmonisation across studies, accelerometer models and populations [44].

However, our study did have some limitations. The high proportion of missing accelerometer data suggests potential issues with data quality in this population, with 6% participants forgetting to put accelerometers back on after a period of non-wear. Although the proportion of data loss is low compared to some studies [19, 78], we attempted to minimise this through choice of wrist-worn accelerometers, which have shown good compliance in children [42]. However, data loss in accelerometer processing, and poor accelerometer wear compliance are ongoing issues among accelerometer studies [78, 79]. Future studies should examine the feasibility and acceptability of wrist-worn accelerometers in this population to minimise missing data. In addition, our sample sizes did not allow for further breakdown by age. Finally, some COVID-19 restrictions were in place in Malaysia during the data collection period. However, this allowed us to explore the effects of remaining COVID-19 restrictions on PA engagement.

## Conclusions

This is the first study exploring 24-hour accelerometer-measured movement behaviours in a population of children and adolescents in South East Asia. We observed low PA engagement, few participants meeting WHO MVPA recommendations, and differences in PA profiles between different age, sex, ethnicity and socioeconomic groups, consistent with previous studies among children in Malaysia and worldwide. However, our findings that being female, Malay ethnicity and an adolescent remained the strongest correlates of low childhood PA engagement lead us to recommend that future studies should seek to understand why these subgroups have particularly low PA levels within Malaysia.

## Supporting information

**S1 Table. Missing data.**
(DOCX)

**S2 Table. Accelerometer-measured physical activity proportions (%/day) split by demographic characteristics.**
(DOCX)

**S3 Table. Participants meeting WHO physical activity guidelines.**
(DOCX)

**S4 Table. Accelerometer-measured physical activity intensity gradient and average acceleration split by demographic characteristic.**
(DOCX)

**S5 Table. Regression models investigating differences in physical activity outcomes between demographic groups.**
(DOCX)

**S6 Table. Sensitivity analysis of regression models investigating differences in physical activity outcomes between demographic groups removing participants with data collection during Ramadan (3rd April– 1st May 2022).**
(DOCX)

**S7 Table. Physical activity levels of participants with data collection during Ramadan (3rd April– 1st May 2022).**
(DOCX)

**S8 Table. Physical activity levels of participants with data collection during COVID-19 related rotating school systems.**
(DOCX)

**S9 Table. Linear regression model results for associations between external events with moderate-to-vigorous physical activity.**
(DOCX)

## Acknowledgments

The authors would like to express their appreciation to the SEACO Field Teams and survey participants. The research described in this paper was supported by the South East Asia Community Observatory (SEACO, https://www.monash.edu.my/seaco). The views, however, are those of the authors and there is no real or implied endorsement by SEACO.

## Author Contributions

**Conceptualization:** Sophia M. Brady, Ruth Salway, Louise Millard, Laura Johnson, Tin Tin Su, Miranda E. G. Armstrong.

**Data curation:** Sophia M. Brady, Ruth Salway.

**Formal analysis:** Sophia M. Brady, Ruth Salway.

**Funding acquisition:** Louise Millard, Andy Skinner, Laura Johnson, Tin Tin Su, Miranda E. G. Armstrong.

**Methodology:** Sophia M. Brady, Ruth Salway, Jeevitha Mariapun, Louise Millard, Amutha Ramadas, Hussein Rizal, Andy Skinner, Chris Stone, Laura Johnson, Tin Tin Su, Miranda E. G. Armstrong.

**Supervision:** Ruth Salway, Miranda E. G. Armstrong.

**Writing – original draft:** Sophia M. Brady.

**Writing – review & editing:** Ruth Salway, Jeevitha Mariapun, Louise Millard, Amutha Ramadas, Hussein Rizal, Andy Skinner, Chris Stone, Laura Johnson, Tin Tin Su, Miranda E. G. Armstrong.

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
