## [Decision Letter · Decision Letter 0]

5 Nov 2023

PONE-D-23-31758Accelerometer-measured 24-hour movement behaviours over 7 days in Malaysian children and adolescents: a cross-sectional studyPLOS ONE

Dear Dr. Armstrong,

Thank you for submitting your manuscript to PLOS ONE. After careful consideration, we feel that it has merit but does not fully meet PLOS ONE’s publication criteria as it currently stands. Therefore, we invite you to submit a revised version of the manuscript that addresses the points raised during the review process.

We look forward to receiving your revised manuscript.

Kind regards,

Zulkarnain Jaafar

Academic Editor

PLOS ONE

Reviewers' comments:

Reviewer's Responses to Questions

**Comments to the Author**

1. Is the manuscript technically sound, and do the data support the conclusions?

Reviewer #1: Yes

Reviewer #2: Yes

2. Has the statistical analysis been performed appropriately and rigorously? 

Reviewer #1: Yes

Reviewer #2: Yes

3. Have the authors made all data underlying the findings in their manuscript fully available?

Reviewer #1: Yes

Reviewer #2: Yes

4. Is the manuscript presented in an intelligible fashion and written in standard English?

Reviewer #1: Yes

Reviewer #2: Yes

5. Review Comments to the Author

Reviewer #1: Firstly, thanks for the opportunity to review this article. The aim of this study is relatively new on this population which makes the purpose a good opportunity for publication, aiming that other researchers may replicated the study and find the same or different results, leading to a strong support in literature.

The article presents a robust and consistent literature review, within the topic and with matched references. The aim of the study is well described. The authors described the materials and methods very clear and concise, and the most relevant information is well established.

Results are very descriptive. Authors showed the most important data from the study that lead to an easy comprehensive conclusion. All results (i.e., tables and figures) the authors were concerned to include in the study.

Lastly, congratulations on the selection of the study topic and good work.

Reviewer #2: Initially, I would like to congratulate the authors for their dedicated work in exploring physical activity and movement over a 24-hour period in Malaysian children and adolescents, using wrist-worn accelerometers in conjunction with self-report questionnaires. The research also deserves recognition for its investigation of differences in time spent in sleep, inactivity, light physical activity (LPA), and moderate-to-vigorous physical activity (MVPA) among different demographic groups and external influences. This approach is particularly significant due to the scarcity of studies addressing this population. The integration of data from questionnaires and accelerometers enhances the accuracy in assessing time spent in physical activity, sleep, and inactivity in this population.

However, I would like to make some suggestions with the aim of improving the study's methodology and providing greater clarity for other researchers interested in replicating these methods, as well as deepening the understanding of practices in the field of accelerometry.

I suggest that the methods section specify which "epoch" was used in the research. This is important because some studies indicate that the choice of "epoch" can impact the recorded time of physical activity, and this information is crucial for data processing and assisting other researchers interested in conducting similar studies.

It would also be relevant to provide information on the accelerometer's sampling rate (in Hz) in the methods, as this information is essential for assisting researchers in selecting the appropriate configuration for other studies.

Regarding the choice of placing the accelerometer on the dominant wrist, it would be enlightening to know whether this decision was made due to greater acceptance by children and adolescents, or for another reason. This information is valuable to better understand the reasons behind methodological decisions and can help identify ways to increase reproducibility in the use of accelerometers.

The emphasis given to the use of GGIR software for data processing can be improved in the work. I suggest that this emphasis can be made by adding the term “GGIR” in the keywords or replacing any term in the keywords that is already present in the title. Another way to highlight GGIR could be to introduce it as a strong point of the research. This can increase the visibility of the study and contribute to dissemination, as this is a type of processing that has been growing in this area.

With these suggestions, I believe that the replication of methods by other researchers can be improved, increasing the comparability of research in the field of accelerometry involving the physical activity of children and adolescents. I hope my comments have been helpful and can contribute to reflections on the theme.

6. PLOS authors have the option to publish the peer review history of their article (what does this mean?). If published, this will include your full peer review and any attached files.

Reviewer #1: No

Reviewer #2: No

---

## [Author Response · Author response to Decision Letter 0]

13 Dec 2023

Response to Reviewers

Submission ID: PONE-D-23-31758

Title: Accelerometer-measured 24-hour movement behaviours over 7 days in Malaysian children and adolescents: a cross-sectional study

Response: We have read and discussed all the reviewers’ comments and hope that we have now addressed the comments to the reviewers satisfaction. We have tried to incorporate the specific comments and changes to improve the scientific content of the manuscript and to clarify possible misinterpretations by future readers. Consequently, we believe that we have improved the overall manuscript. We have tracked all changes and we have highlighted these edits (or text around deleted sections) with a different colour.

Author notes: Changes are listed as page and paragraph taken from the tracked changes version of the revision. The location will have moved in the clean version of the manuscript.

Response: We have checked these style requirements and have updated our formatting and style according to fit within these. 

Response: Data Availability Statement: Data cannot be shared publicly because of confidentiality and ethical reasons. De-identified data are available and can be freely requested from the South East Asia Community Observatory, Monash University Malaysia Institutional Data Access at "mum.seaco@monash.edu" for researchers who meet the criteria for access to confidential data. For more information, please refer to https://www.monash.edu.my/seaco/research-and-training/how-to-collaborate-with-seaco . 

Response: We have checked the reference list, and all references are present and correct. New additions to the reference list are references 42 and 43, in line with additions to the text mentioned below. 

Responses to reviewers’ comments

Reviewer 1

Firstly, thanks for the opportunity to review this article. The aim of this study is relatively new on this population which makes the purpose a good opportunity for publication, aiming that other researchers may replicated the study and find the same or different results, leading to a strong support in literature.

The article presents a robust and consistent literature review, within the topic and with matched references. The aim of the study is well described. The authors described the materials and methods very clear and concise, and the most relevant information is well established.

Results are very descriptive. Authors showed the most important data from the study that lead to an easy comprehensive conclusion. All results (i.e., tables and figures) the authors were concerned to include in the study.

Lastly, congratulations on the selection of the study topic and good work.

Response: We would like to thank Reviewer 1 for these kind comments on our manuscript. We hope our minor changes improve the manuscript further and they will be in a position to recommend the manuscript for publication. 

Reviewer 2

Initially, I would like to congratulate the authors for their dedicated work in exploring physical activity and movement over a 24-hour period in Malaysian children and adolescents, using wrist-worn accelerometers in conjunction with self-report questionnaires. The research also deserves recognition for its investigation of differences in time spent in sleep, inactivity, light physical activity (LPA), and moderate-to-vigorous physical activity (MVPA) among different demographic groups and external influences. This approach is particularly significant due to the scarcity of studies addressing this population. The integration of data from questionnaires and accelerometers enhances the accuracy in assessing time spent in physical activity, sleep, and inactivity in this population.

Response: We would like to thank Reviewer 2 for these comments on the manuscript. We hope that our paper makes a significant contribution to the physical activity research area within Malaysian children and adolescents.

However, I would like to make some suggestions with the aim of improving the study's methodology and providing greater clarity for other researchers interested in replicating these methods, as well as deepening the understanding of practices in the field of accelerometry.

I suggest that the methods section specify which "epoch" was used in the research. This is important because some studies indicate that the choice of "epoch" can impact the recorded time of physical activity, and this information is crucial for data processing and assisting other researchers interested in conducting similar studies.

Response: We thank the reviewer for this perceptive comment. Please see our addition specifying the epoch used in page 7, paragraph 1, line 109-110. 

[“Accelerometers were set up to record data at a frequency of 100Hz, and then the collected data was converted to 5 second epochs.”]

It would also be relevant to provide information on the accelerometer's sampling rate (in Hz) in the methods, as this information is essential for assisting researchers in selecting the appropriate configuration for other studies.

Response: Please see the additions specifying the sampling rate in the methods section (page 7, paragraph 1, line 109-110).

[“Accelerometers were set up to record data at a frequency of 100Hz, and then the collected data was converted to 5 second epochs.”]

Regarding the choice of placing the accelerometer on the dominant wrist, it would be enlightening to know whether this decision was made due to greater acceptance by children and adolescents, or for another reason. This information is valuable to better understand the reasons behind methodological decisions and can help identify ways to increase reproducibility in the use of accelerometers.

Response: We thank the reviewer for this comment. Please see additions in the methods section for justification of why the dominant wrist was chosen as out accelerometer placement site of choice (page 7, paragraph 1, line 105-108).

[“This placement site was chosen due to high acceptability and compliance of wrist-worn accelerometers in children (42), and it being a commonly used placement site in large-scale population-based cohort studies (43), allowing for greater comparability of our results.”]

The emphasis given to the use of GGIR software for data processing can be improved in the work. I suggest that this emphasis can be made by adding the term “GGIR” in the keywords or replacing any term in the keywords that is already present in the title. Another way to highlight GGIR could be to introduce it as a strong point of the research. This can increase the visibility of the study and contribute to dissemination, as this is a type of processing that has been growing in this area.

Response: We agree with the reviewer that a greater emphasis of our use of GGIR would be beneficial. We have added GGIR to the keywords and updated the keywords (see submission) and added a few sentences highlighting the strengths of this methodological approach in the discussion (page 23, paragraph 2, line 396-399). 

[“Lastly, our use of the open-source GGIR package allows for the processing and analysis of raw accelerometer data and facilitates greater reproducibility of analysis. GGIR is not accelerometer brand specific, and its capabilities allow for a greater level of data harmonisation across studies, accelerometer models and populations (44).”]

With these suggestions, I believe that the replication of methods by other researchers can be improved, increasing the comparability of research in the field of accelerometry involving the physical activity of children and adolescents. I hope my comments have been helpful and can contribute to reflections on the theme.

Response: Many thanks to this reviewer for these comments- we hope our changes and greater clarity added in the methodologies and other sections of the manuscript are now sufficient for the reviewer to recommend this manuscript for publication.

---

## [Decision Letter · Decision Letter 1]

27 Dec 2023

Accelerometer-measured 24-hour movement behaviours over 7 days in Malaysian children and adolescents: a cross-sectional study

PONE-D-23-31758R1

Dear Dr. Armstrong,

We’re pleased to inform you that your manuscript has been judged scientifically suitable for publication and will be formally accepted for publication once it meets all outstanding technical requirements.

Kind regards,

Zulkarnain Jaafar

Academic Editor

PLOS ONE

Additional Editor Comments (optional):

Reviewers' comments:

Reviewer's Responses to Questions

**Comments to the Author**

1. If the authors have adequately addressed your comments raised in a previous round of review and you feel that this manuscript is now acceptable for publication, you may indicate that here to bypass the “Comments to the Author” section, enter your conflict of interest statement in the “Confidential to Editor” section, and submit your "Accept" recommendation.

Reviewer #2: All comments have been addressed

2. Is the manuscript technically sound, and do the data support the conclusions?

Reviewer #2: Yes

3. Has the statistical analysis been performed appropriately and rigorously? 

Reviewer #2: Yes

4. Have the authors made all data underlying the findings in their manuscript fully available?

Reviewer #2: Yes

5. Is the manuscript presented in an intelligible fashion and written in standard English?

Reviewer #2: Yes

6. Review Comments to the Author

Reviewer #2: (No Response)

7. PLOS authors have the option to publish the peer review history of their article (what does this mean?). If published, this will include your full peer review and any attached files.

Reviewer #2: No

---

## [Editor Report · Acceptance letter]

9 Feb 2024

PONE-D-23-31758R1 

PLOS ONE

Dear Dr. Armstrong, 

I'm pleased to inform you that your manuscript has been deemed suitable for publication in PLOS ONE. Congratulations! Your manuscript is now being handed over to our production team.

Kind regards, 

on behalf of

Dr. Zulkarnain Jaafar 

Academic Editor

PLOS ONE